# SCAR: SHAPLEY CREDIT ASSIGNMENT FOR MORE EFFICIENT RLHF

## ABSTRACT

Reinforcement Learning from Human Feedback (RLHF) is a widely used technique for aligning Large Language Models (LLMs) with human preferences, yet it often suffers from sparse reward signals, making effective credit assignment challenging. In typical setups, the reward model provides a single scalar score for an entire generated sequence, offering little insight into which token or span-level decisions were responsible for the outcome. To address this, we propose Shapley Credit Assignment Rewards (SCAR), a novel method that leverages Shapley values in cooperative game theory. SCAR distributes the total sequence-level reward among constituent tokens or text spans based on their principled marginal contributions. This creates dense reward signals, crucially, without necessitating the training of auxiliary critique models or recourse to fine-grained human annotations at intermediate generation stages. Unlike prior dense reward methods, SCAR offers a game-theoretic foundation for fair credit attribution. Theoretically, we demonstrate that SCAR preserves the original optimal policy, and empirically, across diverse tasks including sentiment control, text summarization, and instruction tuning, we show that SCAR converges significantly faster and achieves higher final reward scores compared to standard RLHF and attention-based dense reward baselines. Our findings suggest that SCAR provides a more effective and theoretically sound method for credit assignment in RLHF, leading to more efficient alignment of LLMs.

## 1 INTRODUCTION

Large Language Models (LLMs) have demonstrated remarkable capabilities in diverse natural language tasks, yet aligning their outputs with complex human instructions and preferences remains a critical challenge Brown et al. (2020); Bai et al. (2022). Reinforcement Learning from Human Feedback (RLHF) has emerged as a powerful paradigm for this alignment Christiano et al. (2017); Stiennon et al. (2020); Ouyang et al. (2022), enabling models to generate more helpful, harmless, and honest responses. The standard RLHF pipeline typically involves training a reward model (RM) on human preference data and then using this RM to fine-tune the LLM via reinforcement learning algorithms like PPO Schulman et al. (2017).

A fundamental challenge in this pipeline arises from the nature of reward delivery. LLMs generate text auto-regressively, selecting one token at a time. Each token selection constitutes an action within an RL framework. However, the RM typically provides only a single scalar reward signal upon the completion of the entire sequence Stiennon et al. (2020); Bai et al. (2022). This results in a sparse reward setting Hare (2019), which is notoriously difficult for RL algorithms to optimize efficiently Yu (2018). The core difficulty lies in credit assignment Sutton (1984); Lansdell et al. (2019): determining which specific token choices (actions) early in the generation process were responsible for the final high or low reward. This sparsity can lead to slow convergence, high variance gradients, training instability Henderson et al. (2018); Razin et al. (2024), and potentially suboptimal final policies.

Recognizing this limitation, recent work has focused on densifying the reward signal specifically for RLHF. One approach involves training dense reward models using fine-grained human annotations at intermediate generation steps Lightman et al. (2023); Wu et al. (2023b). While effective, this approach incurs substantial annotation costs. To mitigate these costs, auxiliary LLMs have been proposed as critics to generate intermediate feedback Klissarov et al. (2024); Cao et al. (2024); Yoon et al.

(2024), though this relies on the critic's inherent capabilities. An alternative strategy, most relevant to our work, aims to derive dense rewards by redistributing the existing sequence-level reward from the RM without new annotations or auxiliary models. A notable example is Attention-Based Credit (ABC) Chan et al. (2024), which repurposes the RM's final layer's attention weights to proportionally distribute the final reward across tokens. ABC provides a token-level signal at minimal computational overhead while preserving optimality via potential-based reward shaping Ng et al. (1999b).

While ABC offers a practical heuristic, relying on attention weights for credit assignment presents several limitations. Firstly, attention mechanisms are not explicitly designed for, nor do they offer a theoretically grounded guarantee of, fair credit attribution; their interpretability for attribution is debated Jain & Wallace (2019); Wiegreffe & Pinter (2019). Secondly, relying on the final layer's attention is insufficient, as significant amount of information extraction and reasoning processes that influence the final reward prediction occur throughout the intermediate layers of the model Geva et al. (2022); Meng et al. (2022). Lastly, standard attention scores are non-negative, making it inherently difficult to assign explicit negative credit to tokens or spans that detract from the output quality.

In our pursuit of a more principled approach to dense credit assignment in RLHF, we turn to cooperative game theory and propose Shapley Credit Assignment Rewards (SCAR). The core idea is to model sequence generation as a cooperative game where individual text segments (e.g., tokens, or larger spans like phrases or sentences) are the players. The value of any coalition of these players (i.e., a subset of segments) is determined by querying the reward model with the partial sequence formed by concatenating the segments in that coalition. We then employs the Shapley value (Shapley, 1953) to fairly distribute the total sequence-level reward among these segments. This distribution is based on each segment's average marginal contribution across all possible orderings of player arrivals, offering a theoretically grounded method that uniquely satisfies desirable axioms such as *efficiency, symmetry*, and *linearity*, thereby ensuring a fair and principled allocation of credit Shapley (1953).

Recognizing that the exact computation of Shapley values is exponentially complex in the number of players and thus prohibitive for long sequences, we incorporate adaptive text segmentation to maintain a tractable number of players. We leverage constituency parsing to segment the generated text into a hierarchical structure of syntactically coherent units (e.g., phrases or clauses), which then serve as the players. This significantly reduces the computational burden by grouping semantically related tokens. For longer responses, segmentation at the sentence level provides a further practical alternative. We empirically validate SCAR across three NLP tasks with different response length, demonstrating that it consistently leads to faster convergence and achieves higher final reward scores compared to standard sparse RLHF and attention-based dense reward baselines.[1]

## 2  RELATED WORK

### 2.1  RL FOR TEXT GENERATION

Reinforcement Learning (RL) has been increasingly leveraged for fine-tuning LLMs in text generation tasks Ryang & Abekawa (2012); Norouzi et al. (2016); Li et al. (2016); Buck et al. (2018); Bahdanau et al. (2017). Unlike standard supervised fine-tuning which relies on maximizing the likelihood of ground-truth sequences, RL enables optimization directly towards sequence-level objectives or metrics that are not differentiable, such as ROUGE scores in summarization or human preferences for dialogue quality (Stiennon et al., 2020; Ouyang et al., 2022). A particularly successful application of this is Reinforcement Learning from Human Feedback (RLHF) Ouyang et al. (2022); Christiano et al. (2017); Stiennon et al. (2020); Bai et al. (2022), which has become a widely adopted technique for aligning LLMs with complex human values and instructions. The typical RLHF process involves learning a reward model (RM) from a dataset of human comparisons between different model outputs, followed by fine-tuning the LLM policy to maximize the scalar reward assigned by the RM to the generated text sequences. However, a widely acknowledged challenge in this standard RLHF framework is the inherent sparsity of the reward signal: the RM typically provides feedback only after the entire sequence has been generated Stiennon et al. (2020); Bai et al. (2022). This terminal reward makes the temporal credit assignment problem—identifying which specific token choices (actions) contributed positively or negatively to the final outcome—particularly difficult Sutton (1984). This

---

[1]Code will be released upon acceptance.

difficulty has spurred research into methods for creating denser, more informative reward signals, which we discuss further in the following subsection.

## 2.2 DENSE REWARD STRATEGIES

Ng et al. (1999a) laid the groundwork for potential-based reward shaping in RL, demonstrating that such shaping can effectively reduce training time without changing the optimal policy. This concept has inspired subsequent research in augmenting learning processes through auxiliary rewards. Bellemare et al. (2016); Gong & Wang (2020); Ostrovski et al. (2017); Tang et al. (2017) have employed pseudo-count-based rewards to encourage exploration in environments where rewards are sparse. Pathak et al. (2017) use the agent's prediction errors as intrinsic reward signals to encourage exploration. Zheng et al. (2018) proposed a method where a parameterized intrinsic reward model is learned during training to generate dense reward signals. This approach, however, presents certain optimization difficulties due to the necessity of calculating second-order gradients. Wu et al. (2023b); Lightman et al. (2023) employ human annotators to provide detailed span-level reward signals, demonstrating that these fine-grained rewards yield better performance compared to holistic rewards. More recently, Attention Based Credit (ABC) (Chan et al., 2024) proposed using the reward model's internal attention weights to redistribute the terminal reward across tokens. ABC provides density "for free" and preserves optimality via potential-based reward shaping (PBRS) (Ng et al., 1999a). However, attention weights are a heuristic proxy for importance and may not reflect a principled allocation of credit. Our method differs fundamentally by employing Shapley Values, a concept grounded in cooperative game theory, to achieve a fair and principled distribution of the reward based on the marginal contribution of text segments.

## 2.3 SHAPLEY VALUES IN LLMS

Several previous works have already proposed approaches to interpreting the behaviour of LLMs using Shapley values. Goldshmidt & Horovicz (2024) enhanced LLM output interpretability by calculating the Shapley values of sub-sequences. Mohammadi (2024) demonstrated how the Shapley value can uncover that the LLM decisions are disproportionately influenced by tokens providing minimal informative content. Liu et al. (2023) used Shapley values to quantify the value of prompts equitably in multi-prompt learning methods. In addition to enhancing the interpretability of LLM outputs, Shapley value methods are also used for model pruning/compression Sun et al. (2025) and dataset refinement He et al. (2024). However, to the best of our knowledge, there's limited work on solving the sparse reward problem in RLHF using Shapley values. We note contemporaneous work by Koo et al. (2025) which also explores Shapley values for reward distribution. Their approach estimates per-token rewards using interpretability methods and then employs Bayesian Optimization (BO) in a bilevel framework to learn parameters for a shaping function that combines these estimates. Our work, SCAR, differs in two ways. First, to address the inefficiency of token-level Shapley calculations for longer responses, SCAR allows for adaptive text segmentation, including grouping tokens into larger syntactic spans (e.g., via constituency parsing) for more efficient credit assignment. Secondly, unlike Koo et al. (2025) requiring an outer optimization loop (like BO) to learn shaping parameters, SCAR directly uses the (approximated) Shapley values from these text units to construct the dense reward signal (Equation 6).

## 3 SHAPLEY CREDIT ASSIGNMENT REWARDS

This section details our proposed method, Shapley Credit Assignment Rewards (SCAR). We first review the standard RLHF setup and the inherent reward sparsity problem. We then introduce a game-theoretic perspective on text generation for credit assignment, define SCAR based on Shapley values, and finally discuss efficient approximation techniques.

### 3.1 RLHF AND REWARD SPARSITY

We formulate the autoregressive text generation process using the standard Markov Decision Process (MDP) formalism, denoted by $\mathcal{M} = (\mathcal{S}, \mathcal{A}, P, R, \gamma)$. The process begins in an initial state $s_0 \in \mathcal{S}$, which corresponds to the input prompt $x$. At each discrete time step $t \in \{0, 1, \ldots, T-1\}$, the agent (the language model) is in state $s_t$, representing the concatenation of the prompt and the

sequence generated thus far: $s_t = x \oplus y_{1:t}$ (where $y_{1:0}$ denotes an empty sequence, so $s_0 = x$, and $\oplus$ denotes token concatenation). The agent selects an action $a_t \in \mathcal{A}$, which corresponds to choosing the next token $y_{t+1}$ from the vocabulary $\mathcal{V}$ according to its policy $\pi_\theta(a_t|s_t)$, parameterized by $\theta$. We identify the action space with the vocabulary, i.e., $\mathcal{A} = \mathcal{V}$. The state transition function $P(s_{t+1}|s_t, a_t)$ is deterministic in this context. Upon taking action $a_t = y_{t+1}$ in state $s_t$, the system transitions to next state $s_{t+1} = s_t \oplus a_t = x \oplus y_{1:t+1}$. This generation process continues until a maximum sequence length $T$ is reached or a designated end-of-sequence (EOS) token is generated. For notational simplicity, we often assume a fixed horizon $T$. The discount factor $\gamma$ is typically set to 1 in finite-horizon text generation tasks.

In standard RLHF pipeline (Christiano et al., 2017; Stiennon et al., 2020; Ouyang et al., 2022), a reward model $R_\phi$, parameterized by $\phi$, is trained beforehand on a dataset of human preferences $\mathcal{D}_{\text{pref}} = \{(y^w, y^l)_i\}$, where $y^w$ is preferred over $y^l$ by human annotators. The reward model assigns a scalar score $r_\phi(x, y)$ reflecting the quality or preference level of a *completed* sequence $y$ given the prompt $x$.

To stabilize training and prevent the policy $\pi_\theta$ from drifting too far from a reference distribution (often the initial pre-trained LLM, denoted $\pi_{\text{ref}}$), the reward signal used for optimization is typically augmented with a penalty term at each step $t$. A common choice is the Kullback-Leibler (KL) divergence between the current policy $\pi_\theta$ and the reference policy $\pi_{\text{ref}}$. The standard objective is often formulated as:

$$\mathcal{J}(\theta) = \mathbb{E}_{x \sim \mathcal{D}, y \sim \pi_\theta} \left[ \sum_{t=1}^{T} R_t^{\text{orig}} \right] \tag{1}$$

where $\mathcal{D}$ is the dataset and the reward at each timestep $t$ in the standard sparse setup is given by

$$R_t^{\text{orig}} = R_t^{\text{KL}} + \mathbb{I}(t = T) \cdot r_\phi(x, y) \tag{2}$$

Here, $R_t^{\text{KL}} = -\beta \log(\pi_\theta(y_t|x, y_{<t})/\pi_{\text{ref}}(y_t|x, y_{<t}))$ is the KL penalty associated with timestep $t$, $\beta$ is the KL coefficient, and $\mathbb{I}(t = T)$ is an indicator function ensuring the reward model score $r_\phi(x, y)$ is assigned only at the final timestep $T$. This terminal assignment makes the reward signal inherently *sparse*, posing a significant challenge for credit assignment during RL training. Such sparsity directly undermines the efficiency of the learning process and frequently leads to the convergence to suboptimal policies (Ng et al., 1999a; Bellemare et al., 2016; Wu et al., 2023a). To overcome this, a principled approach to redistribute the terminal reward more densely across the generation steps is desirable.

## 3.2 A Game-Theoretic Framework for Credit Assignment

We frame the generation of a sequence $y$ for a given prompt $x$ as a cooperative game. Let the generated text $y$ be segmented into $N$ contiguous units, $y = (u_1, u_2, \ldots, u_N)$. These units could be tokens, spans, sentences, or paragraphs depending on the task. The "players" in this game are these $N$ text units. Let $\mathcal{P} = \{u_1, \ldots, u_N\}$ be the set of players.

**Characteristic Function.** The value of cooperation among a subset (coalition) $S \subseteq \mathcal{P}$ of players is defined by a characteristic function $v : 2^{\mathcal{P}} \to \mathbb{R}$. This function should quantify the collective contribution of the units in $S$ towards the final reward objective. We define $v(S)$ based on the reward model's evaluation of the partial text sequence formed by concatenating the units $\{u_i \mid u_i \in S\}$ in their original order. Let $y_S$ denote this concatenated partial sequence. Then, the value function is defined as:

$$v(S) = r_\phi(x, y_S) \tag{3}$$

For the empty set, $v(\emptyset) = 0$. The value of the grand coalition $v(\mathcal{P})$ corresponds to the original sparse reward for the complete sequence, $v(\mathcal{P}) = r_\phi(x, y)$. Note that evaluating $r_\phi$ on partial sequences $y_S$ requires careful consideration, as $y_S$ represents an incomplete sequence. Ideally, $v(S)$ could represent the expected reward obtained by keeping the units in $S$ fixed and sampling the remaining units from the current policy $\pi_\theta$. In our implementation, we resort to a practical approximation. To evaluate $v(S)$, we construct a sequence in which the tokens belonging to units $u_i \in S$ are placed in their original order. The positions corresponding to units $u_j \notin S$ are filled using empty spaces.

**Shapley Value Calculation.** The Shapley Value $\text{SV}_{u_i}(v)$ for a player $u_i \in \mathcal{P}$ (text unit $u_i$) quantifies its fair contribution to the grand coalition value $v(\mathcal{P})$, calculated as the average marginal contribution of player $u_i$ over all possible permutations of player arrivals:

$$\text{SV}_{u_i}(v) = \sum_{S \subseteq \mathcal{P} \backslash \{u_i\}} \frac{|S|!\,(N - |S| - 1)!}{N!} [v(S \cup \{u_i\}) - v(S)]. \tag{4}$$

The Shapley values uniquely satisfies axioms such as *efficiency* ($\sum_{u_i \in \mathcal{P}} \text{SV}_{u_i}(v) = v(\mathcal{P})$), *symmetry* (equal reward for equal contribution), *linearity*, and the *null player* property (no contribution means no credit), making it a principled choice for fair credit allocation (Shapley, 1953).

### 3.3 Shapley Values as Dense Rewards

We use the calculated Shapley values to define a dense reward signal for the RL agent. Let unit $u_i$ consist of tokens generated from timestep $t_{i-1} + 1$ up to and including timestep $t_i$ (with $t_0 = 0$, so $t_i$ marks the completion timestep of unit $u_i$). We assign the Shapley value $\text{SV}_{u_i}(v)$ associated with unit $u_i$ as an additional reward component specifically at the timestep $t_i$ marking the completion of that unit. Let $R_t^{\text{shap}}$ denote this Shapley-based reward at timestep $t$. Then,

$$R_t^{\text{shap}} = \begin{cases} \text{SV}_{u_i}(v) & \text{if } t = t_i \text{ for some unit } u_i \\ 0 & \text{otherwise} \end{cases} \tag{5}$$

This component distributes the total reward $r_\phi(x, y)$ across the episode based on the Shapley contributions, since $\sum_{t=1}^{T} R_t^{\text{shap}} = \sum_{i=1}^{N} \text{SV}_{u_i}(v) = r_\phi(x, y)$ (due to the efficiency property, where $N$ is the total number of units).

We then define the total reward $R_t$ provided to the RL agent at timestep $t$ as a convex combination of the dense Shapley-based credit allocation and the original sparse terminal reward, while retaining the per-step KL penalty. Using a hyperparameter $\alpha \in [0, 1]$, the total reward is:

$$R_t(\alpha) = R_t^{\text{KL}} + \alpha \cdot R_t^{\text{shap}} + (1 - \alpha) \cdot \mathbb{I}(t = T) \cdot r_\phi(x, y) \tag{6}$$

Here, $\alpha$ controls the interpolation:

- If $\alpha = 0$, $R_t(0) = R_t^{\text{orig}}$, recovering the standard sparse reward signal.
- If $\alpha = 1$, $R_t(1) = R_t^{\text{KL}} + R_t^{\text{shap}}$. The terminal reward $r_\phi(x, y)$ is entirely replaced by an equivalent value distributed densely according to Shapley contributions throughout the episode.
- If $0 < \alpha < 1$, the agent receives both the intermediate Shapley-based rewards and a residual portion of the original terminal reward.

This formulation allows flexible control over the density of the reward signal, balancing immediate feedback with the final outcome signal.

**Theorem 3.1** (Policy Invariance under SCAR Reward Shaping). *Consider a parameterized language model $\pi_\theta$ with a learned reward model $R_\phi$. Let $\mathcal{M} = (\mathcal{S}, \mathcal{A}, P, R_t^{orig}, \gamma)$ be the original MDP with its reward from the reward model and $\widehat{\mathcal{M}} = (\mathcal{S}, \mathcal{A}, P, R_t(\alpha), \gamma)$ be the MDP with dense Shapley reward. If $\pi_\theta$ is optimal for $\widehat{\mathcal{M}}$, then $\pi_\theta$ is also optimal for $\mathcal{M}$, and vice versa.*

*Proof.* See Appendix B. $\qquad\qquad\qquad\qquad\qquad\qquad\qquad\qquad\qquad\qquad\qquad\qquad\square$

### 3.4 Efficient Approximation of Shapley Values

The direct calculation of Shapley values using Equation equation 4 necessitates evaluating the characteristic function $v(S)$ (defined in Eq. equation 3) for all $2^N$ possible coalitions $S$ of the $N$ text units. This exponential complexity renders exact computation practically infeasible for typical sequence lengths encountered in text generation Shapley (1953). To make SCAR practical, we employ two key strategies: adaptive segmentation of text into units and efficient approximation of their Shapley values.

**Adaptive Text Segmentation as Players.** The definition of "players" (text units $u_i$) in our cooperative game is crucial for both interpretability and computational tractability. We adapt the granularity of these units based on the task and the typical length of the generated responses, aiming to keep the number of players $N$ manageable. We experiment with three levels of segmentation:

- **Token-level:** For tasks producing very short responses, each token $y_t$ can be treated as an individual player $u_i$. This offers the finest granularity but results in a larger $N$.

- **Span-level:** For medium-length responses, we leverage constituency parsing (Marcus et al., 1993) to establish a hierarchical grammatical structure over the generated sequence $y$. This process yields a constituency tree where tokens (leaf nodes) are organized into hierarchically nested constituents. Players are then defined as these syntactic constituents (e.g., noun phrases, verb phrases), formed by grouping tokens that share a common parent or ancestor node within this tree. This approach reduces $N$ while preserving semantic coherence within each player unit, as constituents are inherently meaningful linguistic units.[2]

- **Sentence-level:** For tasks generating longer, multi-sentence responses, each sentence in the output $y$ constitutes a player. Segmentation is achieved using standard sentence boundary detection. This approach markedly reduces $N$, especially for verbose outputs.

The choice of segmentation strategy is a hyperparameter, allowing a trade-off between the granularity of credit assignment and the computational cost of Shapley value estimation.

**Approximation Using Owen Values.** To ensure the practical applicability of SCAR, we employ an approximation scheme based on Owen value (Owen, 1977), which is a coalitional extension of Shapley values designed for games where players are grouped into a predefined coalition structure(Aumann & Dreze, 1974). For our task, a hierarchical structure $\mathcal{B}$ is imposed on the sequence of $N$ text units, achieved by applying a heuristic parsing algorithm to the units. This partition $\mathcal{B}$ defines nested groupings of the units. The Owen value is then computed with respect to this partition $\mathcal{B}$. Marginal contributions are evaluated by forming coalitions structurally: combinations involving subsets within a unit's own group are explored, while units belonging to other groups in the partition are treated as indivisible blocks, as they are either entirely included or entirely excluded from a coalition, rather than exploring all their individual subsets. By limiting the evaluation to coalitions dictated by the partition structure $\mathcal{B}$, the number of required characteristic function evaluations (reward model queries) is substantially reduced compared to the exact Shapley computation. Consequently, the computational complexity is reduced from exponential, $O(2^N)$, to quadratic in $N$, rendering the approach tractable. We use the SHAP package Lundberg & Lee (2017) for Shapley values and Owen values computation.

## 4 EXPERIMENTS

In this section, we empirically evaluate the effectiveness of SCAR across three distinct tasks characterized by varying response lengths. Our primary objective is to demonstrate that SCAR enables more efficient and effective training compared to standard sparse RLHF and alternative dense reward baselines.

### 4.1 EXPERIMENTAL SETUP

**Evaluation Tasks and Models.** We evaluate our proposed method across three diverse tasks prevalent in RLHF research: sentiment control, text summarization, and instruction tuning (Chan et al., 2024; Cao et al., 2024; Yoon et al., 2024). For sentiment control and instruction tuning, we utilize the implementation from Chan et al. (2024). However, for summarization, due to difficulties in reproducing the results, we switched to the implementation from Huang et al. (2024). The datasets, policy models, and reward models used for each task are described in more detail below.

**Sentiment Control:** The objective is to generate positive reviews of movies. We use the IMDB dataset (Maas et al., 2011). The policy model is GPT-2 small (Radford et al., 2019), initialized by fine-tuning for one epoch on the IMDB training set. During RLHF training, prompts are generated by

---

[2]https://www.nltk.org/howto/parse.html

| **Prompt:** | "While some scenes were" |
| --- | --- |

**Sparse:**
initially disturbing to sit through, they ultimately contributed to a deeply powerful
0.0    0.0    0.0   0.0   0.0   0.0   0.0     0.0    0.0 0.0   0.0    0.0
and moving story . <EOS>
0.0   0.0   0.0 0.0 +10.0

**ABC:**
initially disturbing to sit through, they ultimately contributed to a deeply
0.0    0.0    0.0 0.0   0.0   0.0   0.0     0.0    0.0 0.0 +0.8
powerful and moving story . <EOS>
+0.8 +0.2 +1.3 +1.6 +5.3 0.0

**Ours:**
initially disturbing to sit through, they ultimately contributed to a deeply
+0.8 −0.5 −0.1 −1.8 −0.8 −0.2 +1.0 +0.7 0.0 +0.5 +1.8
powerful and moving story . <EOS>
+2.2 +0.5 +4.0 +0.7 +0.8 0.0

Figure 1: Comparison of reward distribution strategies for an example generated sequence. Sparse RLHF assigns the total reward at the end. SCAR and ABC distribute this reward across tokens/spans based on their respective methodologies, shown with background highlights (color hue for sign, intensity for magnitude; more intense/saturated means higher absolute contribution) and numerical scores.

randomly selecting the first 4 to 8 tokens from reviews in the training data. The reward signals are provided using a pre-trained sentiment classifier, same as (Chan et al., 2024).

**Text Summarization:** We evaluate our method on the automatic text summarization task, following prior work (Stiennon et al., 2020; Chan et al., 2024; Lee et al., 2023). For this evaluation, we use the Reddit TL;DR dataset (V"olske et al., 2017), specifically the filtered version provided by Stiennon et al. (2020), which includes approximately 116K training examples, 6K validation examples, and 6K test examples. Our policy model is Pythia-1B (Biderman et al., 2023), which we initialize via supervised fine-tuning on the training set for 2,000 steps with a batch size of 64. Additionally, we train a 1B-parameter reward model (initialized using the SFT model) on 92K human preference pairs, achieving approximately 74% accuracy on the validation set.

**Instruction Turning:** We evaluate models on the task of following user instructions. To do this, we fine-tune language models using the helpfulness subset of the Anthropic Helpful and Harmless (HH) dataset (Bai et al., 2022), which contains 43K human-written prompts paired with model responses that have been ranked by human annotators for helpfulness. Preference is based on which response is more informative and helpful for the task. The policy model is initialized using the OpenLLaMA-7B model (Geng & Liu, 2023), an open-source reproduction of Meta's LLaMA collection (Touvron et al., 2023) trained on fully open-source dataset. For the reward model, we use the 3B reward model provided by Dong et al. (2023). This reward model was trained using the same HH dataset, where it learns to assign a scores to candidate completions based on their predicted usefulness.

**Baselines.** We compare our proposed method against three key baselines. **RLHF** represents the standard approach using the sparse terminal reward (Eq. 2) with KL regularization. **ABC (Attention Based Credit)** (Chan et al., 2024) uses reward model attention scores for dense rewards distribution. **Uniform** is a baseline that distributes the terminal reward evenly across all tokens. For fair comparison, all methods are optimized using the PPO objective (Schulman et al., 2017) with consistent hyperparameters, detailed in Appendix D. All methods initialize their policy models using the same SFT checkpoint to ensure a common starting point. All experiments were conducted on a single A100 GPU (80GB VRAM), and results are averaged over 5 random seeds.

**Evaluation Metrics.** We track the average reward $r_\phi(x, y)$ per episode during training to evaluate learning speed and the level of convergence. The final performance is reported as the mean reward on the test set after convergence. For the summarization task, we additionally employ **LLM-as-a-judge** (Zheng et al., 2023) evaluation to compare the quality (e.g., accuracy, coverage, conciseness, clarity, and coherence) of summaries generated by models trained with different methods. We randomly sampled 1K summaries from the TL;DR test set for LLM evaluation. To mitigate potential positional

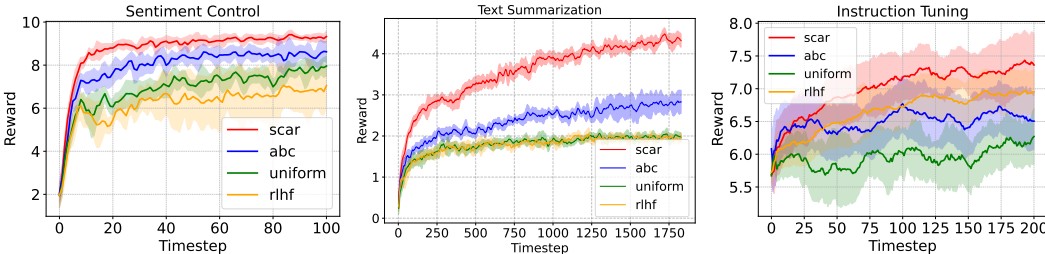

Figure 2: Average reward per timestep during RLHF training for sentiment control (left), text summarization (center), and instruction tuning (right). Curves show the mean reward across five random seeds, with shaded regions representing the standard deviation. SCAR consistently demonstrates faster convergence and achieves higher or comparable final reward levels compared to sparse RLHF, Uniform reward distribution, and Attention-Based Credit (ABC) baselines.

bias in these pairwise comparisons, we randomize the presentation order of summaries. For the instruction-tuning task, we use AlpacaEval (Li et al., 2023) to compare the quality of 1K model's response. AlpacaEval is designed to better handle potential issues such as length bias, thereby providing a more reliable assessment of response quality. The prompt used for evaluation can be found in the Appendix E.

## 4.2 RESULTS

Figure 1 provides a qualitative illustration of how SCAR distributes rewards compared to sparse RLHF and ABC for an example generated sequence. Sparse RLHF, by definition, assigns the entire reward only at the end of the sequence. ABC, which uses attention scores from the reward model's final layer to distribute rewards, tends to concentrate rewards on tokens near the end of the sequence. As seen in the example, significant credit is assigned to the final punctuation mark ("."), while earlier, crucial tokens receive almost zero attention scores. Furthermore, standard attention scores are non-negative, making it difficult for ABC to assign explicit negative credit to tokens or spans that detract from the output quality. For instance, a phrase like "*disturbing to sit through*" that negatively impact the perceived sentiment, would not receive negative rewards from ABC. In contrast, SCAR can assign both positive and negative rewards to tokens based on their game-theoretic marginal contribution.

| Task | Sparse RLHF | Uniform | ABC | SCAR (ours) |
|---|---|---|---|---|
| IMDB | $6.86 \pm 0.86$ | $7.73 \pm 0.02$ | $8.48 \pm 1.60$ | $\mathbf{9.27 \pm 0.00}$ |
| TL;DR | $1.60 \pm 0.11$ | $1.68 \pm 0.02$ | $2.85 \pm 0.21$ | $\mathbf{4.35 \pm 0.11}$ |
| HH-RLHF | $6.93 \pm 0.00$ | $6.17 \pm 0.00$ | $6.59 \pm 0.01$ | $\mathbf{7.31 \pm 0.01}$ |

Table 1: Average reward scores for the trained policy on the test sets for sentiment control (IMDB), text summarization (TL;DR), and instruction tuning (Anthropic HH). Higher scores indicate better performance. Results are averaged over 5 random seeds. Best performance per task is in **bold**.

As depicted in Figure 2, SCAR consistently demonstrated advantages over three baseline methods in terms of learning speed and convergence across all three tasks. This consistent pattern across diverse tasks suggests that the principled, Shapley value-based credit assignment offered by our method effectively improves learning efficiency and enhances the final policy performance. Table 1 presents the average reward scores achieved by each method on the held-out test sets for the three tasks. As shown in the table, SCAR-tuned policy consistently achieves the highest performance compared to the baselines.

| Baselines | Win (%) |
|---|---|
| *Text Summarization (Reddit TL;DR)* | |
| vs. RLHF | 61.2% |
| vs. ABC | 60.3% |
| *Instruction Tuning (Anthropic HH)* | |
| vs. RLHF | 56.3% |
| vs. ABC | 54.9% |

Table 2: LLM-as-Judge pairwise win rates for SCAR against baselines.

For summarization, we use `gemini-2.5-pro` to compare anonymized model outputs (SCAR vs. baselines) on coherence, relevance, conciseness, and overall quality (evaluation prompt in Appendix E). As shown in Table 2, summaries generated by the SCAR-tuned model were preferred over those from the ABC-tuned model in 60.3% and over the sparse RLHFC-tuned model in 61.2%. For the instruction tuning task, we leveraged AlpacaEval (Li et al., 2023) and used `gpt-4-turbo` as an LLM judge, to conduct robust pairwise evaluations. These evaluations assessed helpfulness, harmlessness, and adherence to instructions. SCAR-generated responses achieved a win rate of 54.9% when compared against ABC, and a win rate of 56.3% against standard sparse RLHF. These results provide further evidence that the improvements from SCAR translate to genuinely higher-quality outputs according to human-like preferences.

## 4.3 ANALYSIS

**Reward-KL Tradeoff.** A crucial aspect of RLHF is managing the trade-off between maximizing the reward signal and maintaining proximity to a reference policy ($\pi_{\text{ref}}$). This is often controlled by a KL-divergence penalty term in the reward function (Eq. 2), preventing the policy from drifting too far and generating undesirable or out-of-distribution text, or "reward hacking". Figure 3 illustrates this trade-off by plotting the achieved reward against the KL-divergence from the reference policy $D_{\text{KL}}(\pi \,\|\, \pi_{\text{ref}})$. The plot demonstrates that our method achieves a more favorable reward-KL frontier compared to the standard sparse RLHF baseline. This improved frontier suggests that the principled credit assignment from SCAR en-

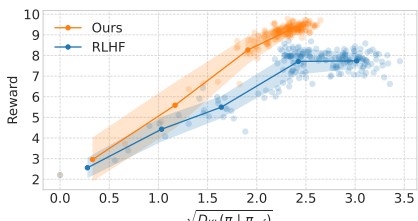

Figure 3: Reward-KL tradeoff on the sentiment control (IMDB) task. The y-axis represents the average per-batch reward during training and the x-axis shows the square root of the KL divergence between the learned policy ($\pi$) and the reference policy ($\pi_{\text{ref}}$).

ables the improvements are genuine and do not come at the cost of generating out-of-distribution or gibberish text.

**Computational Efficiency.** Calculating Shapley values introduces computational overhead compared to sparse RLHF or ABC. Figure 4 plots reward against GPU hours on the TL;DR task. It shows that token-level SCAR, due to its per-step cost, lag behind ABC in reward per GPU hour. However, span-level SCAR demonstrates a much better efficiency profile. Despite the inherent cost of Shapley calculations, its superior sample efficiency allows span-level SCAR to achieve higher reward levels than baselines within the same GPU time. This highlights that by managing the per-step cost through effective segmentation, SCAR's principled dense rewards can lead to greater overall training efficiency.

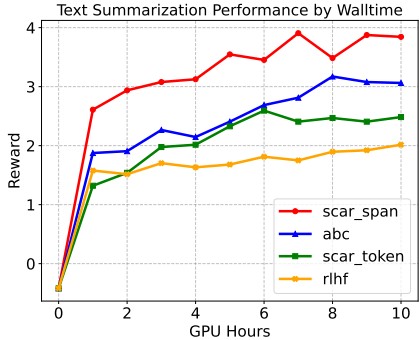

Figure 4: Rewards/GPU hours curves on the TL;DR dataset. We sampled one run from each method. The y-axis represents the reward and the x-axis shows the GPU hours used for training.

## 5 CONCLUSION

We introduced Shapley Credit Assignment Rewards (SCAR), a novel method to address reward sparsity in RLHF by generating dense rewards using Shapley values. Unlike heuristic approaches, SCAR provides a principled, game-theoretic allocation of the reward model's score to text segments. It preserves the optimal policy via potential-based reward shaping and, empirically, demonstrated significantly faster convergence and superior final performance across multiple tasks compared to standard RLHF and other dense reward baselines.

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

## A    LIMITATIONS

Despite its strengths, SCAR has limitations: the computational overhead of Shapley approximations, even with optimizations like Owen values and adaptive segmentation; the assumption that the reward model can meaningfully score partial sequences, which may not suit certain types like rule-based models that only evaluate final answers (e.g., in mathematical reasoning). Future work will target more efficient approximation techniques, robust and adaptive segmentation methods, and rigorous evaluation on larger-scale language models and broader tasks.

## B    OPTIMALITY PRESERVATION

We demonstrate that optimizing with the dense reward signal $R_t(\alpha)$ (defined in Eq. 6) leads to the same optimal policy as optimizing with the original sparse reward $R_t^{\text{orig}}$ (defined in Eq. 2). We leverage the principles of Potential-Based Reward Shaping (PBRS) (Ng et al., 1999a).

**Proof of Theorem 3.1**    According to the theory of potential-based reward shaping (Ng et al., 1999a), if a shaped reward $R'$ differs from an original reward $R$ by a potential-based shaping function $F(s, a, s') = \gamma\Phi(s') - \Phi(s)$, where $\Phi$ is a real-valued function of the state $s$ (the potential function) and $\gamma$ is the discount factor, then the optimal policies remain unchanged.

In our context, the state $s_t$ corresponds to the history $(x, y_{<t})$. The transition $s_t \rightarrow s_{t+1}$ involves selecting action $y_t$. The discount factor $\gamma = 1$. We need to show that the difference between our Shapley values-based reward and the original reward constitutes such a potential function difference.

Let $F_t = R_t(\alpha) - R_t^{\text{orig}}$ be the shaping term added at timestep $t$. Substituting the definitions from Eq. 6 and Eq. 2:

$$F_t = \left( R_t^{\text{KL}} + \alpha \cdot R_t^{\text{shap}} + (1 - \alpha) \cdot \mathbb{I}(t = T) \cdot r_\phi(x, y) \right) - \left( R_t^{\text{KL}} + \mathbb{I}(t = T) \cdot r_\phi(x, y) \right)$$

$$= \alpha R_t^{\text{shap}} + (1 - \alpha)\mathbb{I}(t = T)r_\phi(x, y) - \mathbb{I}(t = T)r_\phi(x, y)$$

$$= \alpha R_t^{\text{shap}} - \alpha\mathbb{I}(t = T)r_\phi(x, y)$$

Let's consider the total undiscounted return for an episode $y = (y_1, \ldots, y_T)$. The total return under the original reward is $G^{\text{orig}} = \sum_{t=1}^{T} R_t^{\text{orig}}$. The total return under the SCAR reward is $G^{\text{shap}}(\alpha) = \sum_{t=1}^{T} R_t(\alpha)$. We have:

$$G^{\text{shap}}(\alpha) = \sum_{t=1}^{T} R_t(\alpha) = \sum_{t=1}^{T}(R_t^{\text{orig}} + F_t)$$

$$= \sum_{t=1}^{T} R_t^{\text{orig}} + \sum_{t=1}^{T} F_t$$

$$= G^{\text{orig}} + \sum_{t=1}^{T} \left( \alpha R_t^{\text{shap}} - \alpha\mathbb{I}(t = T)r_\phi(x, y) \right)$$

$$= G^{\text{orig}} + \alpha \left( \sum_{t=1}^{T} R_t^{\text{shap}} \right) - \alpha r_\phi(x, y)$$

By the efficiency property of Shapley Values, $\sum_{i=1}^{M} \text{SV}_i(v) = v(N) = r_\phi(x, y)$. Since $\sum_{t=1}^{T} R_t^{\text{shap}} = \sum_{i=1}^{M} \text{SV}_i(v)$, we have $\sum_{t=1}^{T} R_t^{\text{shap}} = r_\phi(x, y)$. Substituting this back:

$$G^{\text{shap}}(\alpha) = G^{\text{orig}} + \alpha r_\phi(x, y) - \alpha r_\phi(x, y)$$

$$= G^{\text{orig}}$$

The total undiscounted reward accumulated over any complete episode is identical for both the original reward $R_t^{\text{orig}}$ and the SCAR reward $R_t(\alpha)$.

Since the objective in RL is to maximize the expected total return, $J(\pi) = \mathbb{E}_{\tau \sim \pi}[G]$, and we have shown that $G^{\text{shap}}(\alpha) = G^{\text{orig}}$ for any episode (trajectory) $\tau$, it follows that the expected total returns are identical for any policy $\pi$:

$$J^{\text{shap}}(\pi) = \mathbb{E}_{\tau \sim \pi}[G^{\text{shap}}(\alpha)] = \mathbb{E}_{\tau \sim \pi}[G^{\text{orig}}] = J^{\text{orig}}(\pi) \tag{7}$$

Because the objectives $J^{\text{shap}}(\pi)$ and $J^{\text{orig}}(\pi)$ are identical for all policies $\pi$, any policy $\pi^*$ that maximizes $J^{\text{shap}}$ must also maximize $J^{\text{orig}}$, and vice versa.

Therefore, optimizing the policy $\pi_\theta$ using the dense reward $R_t(\alpha)$ is equivalent to optimizing using the original sparse reward $R_t^{\text{orig}}$ in terms of the resulting optimal policy. This ensures that the benefits of denser rewards provided by Shapley values reshaping (e.g., faster convergence, improved stability) do not come at the cost of altering the fundamental goal of the optimization.

## C  ADDITIONAL EXPERIMENTAL RESULTS

In this section, we present additional experiments that could not be included in the main text due to space limitations.

### C.1  TOKEN-LEVEL VS SPAN-LEVEL SCAR

For the text summarization task, we implemented and compared token-level and span-level SCAR. As illustrated by their reward curves in Figure 5, span-level SCAR achieved performance comparable to that of token-level SCAR. However, span-level SCAR demonstrated significantly greater computational efficiency: on our training platform (one A100 GPU with 80GB VRAM) using the Pythia-1b model, 1K training steps required 48 GPU hours for token-level SCAR versus only 7 GPU hours for span-level SCAR. This makes span-level SCAR approximately seven times more efficient in this specific task (a more detailed time consumption analysis is provided in Figure 4). Due to the high computational cost of token-level SCAR, a statistical study involving multiple runs was infeasible for this method. Consequently, Figure 5 presents a single representative run for each approach to illustrate their comparative performance.

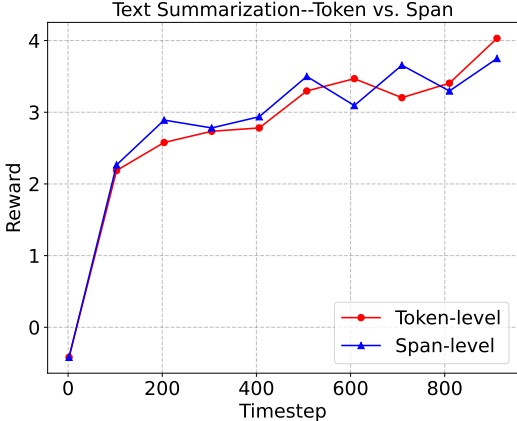

Figure 5: Comparison between token-level and span-level SCAR on the text summarization (TL;DR) task. The y-axis represents the reward during training, and the x-axis shows the training timestep.

# D TABLE OF HYPERPARAMETERS

To ensure experimental reproducibility, this section details the hyperparameters employed. Our Proximal Policy Optimization (PPO) implementation for sentiment control and instruction tuning experiments is based on the Hugging Face TRL library[3] and code from Chan et al. (2024). For the text summarization task, our implementation is built upon the codebase available at `https://github.com/vwxyzjn/summarize_from_feedback_details`.

| Hyperparameter | Value |
|---|---|
| **PPO Hyperparameters** | |
| Learning rate | 1.41e-5 |
| Batch size | 128 |
| Mini-batch size | 128 |
| Gradient accumulation steps | 1 |
| KL coefficient ($\beta$) | 0.2 |
| Shapley value coefficient ($\alpha$) | 0.8 |
| Discount factor ($\gamma$) | 1 |
| Clip range | 0.2 |
| **LLM Hyperparameters** | |
| Min generation length | 16 |
| Max generation length | 24 |
| Temperature | 0.7 |
| Top-p | 1.0 |

Table 3: Hyperparameters used in the sentiment control task.

| Hyperparameter | Value |
|---|---|
| **PPO Hyperparameters** | |
| Learning rate | 3e-6 |
| Batch size | 64 |
| Mini-batch size | 16 |
| Gradient accumulation step | 4 |
| KL coefficient ($\beta$) | 0.05 |
| Shapley value coefficient ($\alpha$) | 1.0 |
| Over-length sequence reward penalty | -1 |
| Discount factor ($\gamma$) | 1 |
| Clip range | 0.2 |
| **LLM Hyperparameters** | |
| Generation length | 53 |
| Temperature | 0.7 |
| Top-p | 1.0 |

Table 4: Hyperparameters used in the text summarization task.

---

[3]`https://huggingface.co/docs/trl/main/en/ppo_trainer`

| Hyperparameter | Value |
|---|---|
| **PPO Hyperparameters** | |
| Learning rate | 1.41e-5 |
| Batch size | 16 |
| Mini-batch size | 2 |
| Gradient accumulation step | 8 |
| KL coefficient ($\beta$) | 0.2 |
| Shapley value coefficient ($\alpha$) | 0.8 |
| Discount factor ($\gamma$) | 1 |
| Clip range | 0.2 |
| LoRA rank | 32 |
| LoRA $\alpha$ | 32 |
| LoRA dropout | 0.0 |
| **LLM Hyperparameters** | |
| Min generation length | 8 |
| Max generation length | 256 |
| Temperature | 1.0 |
| Top-p | 1.0 |

Table 5: Hyperparameters used in the instruction tuning task.

## E  EVALUATION PROMPT FOR TEXT SUMMARIZATION

This section presents the prompt used to query the LLM for summarization quality evaluation. To mitigate potential position bias, the baseline summary and the summary generated by our method were randomly assigned to the `summary_A` and `summary_B` placeholders within the prompt.

---

**Human Evaluation Prompt for Text Summarization**

You are an expert human evaluator specializing in text summarization. Your task is to meticulously compare two summaries, "Summary A" and "Summary B," generated from the same "Original Document." Your goal is to determine which summary is of higher quality overall.

Please consider the following criteria in your evaluation:

1. **Accuracy & Faithfulness:**
   - Does the summary accurately represent the main points of the original document?
   - Does it avoid introducing new information or misinterpreting facts from the document (hallucinations)?

2. **Coverage & Comprehensiveness:**
   - Does the summary cover the most important information and key takeaways from the original document?
   - Are there any critical omissions of essential information?

3. **Conciseness & Succinctness:**
   - Is the summary brief and to the point, avoiding unnecessary jargon, redundancy, or overly verbose phrasing, while still capturing essential information?
   - Is it significantly shorter than the original document, as a good summary should be?

4. **Clarity & Readability:**
   - Is the summary well-written, grammatically correct, easy to understand, and fluent?
   - Is the language clear and precise?

5. **Coherence:**

---

- Do the sentences in the summary flow logically? Does it make sense as a standalone piece of text?
- Is there a logical structure to the summary?

**Input:**

**Original Document:**

```
{original_document}
```

**Summary A:**

```
{summary_A}
```

**Summary B:**

```
{summary_B}
```

**Instructions for your response:**

1. **Reasoning:**
   - First, briefly state your understanding of the main purpose or key points of the **Original Document**.
   - Then, provide a step-by-step comparative analysis of Summary A and Summary B based on the criteria listed above (Accuracy, Coverage, Conciseness, Clarity, Coherence).
   - For each criterion, explicitly compare A and B. For instance, "Regarding Accuracy, Summary A does X well, while Summary B struggles with Y..."
   - Point out specific strengths and weaknesses of each summary. You can reference parts of the summaries or the original document if helpful (e.g., "Summary A correctly captures the detail about X from paragraph 2 of the document, whereas Summary B omits this.").

2. **Overall Decision:**
   - After your detailed reasoning, clearly state which summary you believe is better overall and why, making a holistic judgment. If they are of very comparable quality, or if one excels in some areas while the other excels in others making a clear choice difficult, you can indicate that.

**Output Format:**

First, provide your detailed **Reasoning** as described above. Then, on a new line, write "**Overall Decision:**" followed by your overall assessment. Finally, on a separate, new line, output *only* one letter:

- 'A' if Summary A is better.
- 'B' if Summary B is better.
- 'C' if both summaries are of very similar quality (a tie), or if one is not definitively superior to the other across the most important criteria.

Begin your evaluation now.

