# OpenReview forum: "SCAR: Shapley Credit Assignment for More Efficient RLHF"
_ICLR.cc/2026/Conference — Submitted to ICLR 2026_

### Official Review · Reviewer_cak2 · 2025-10-26

**Soundness:** 3
**Presentation:** 3
**Contribution:** 3
**Rating:** 2
**Confidence:** 4

**Summary:**

This paper proposes SCAR (Shapley Credit Assignment Rewards), which applies Shapley values from cooperative game theory to address the sparse reward problem in Reinforcement Learning from Human Feedback (RLHF). The authors model text generation as a cooperative game where text segments (tokens, spans, or sentences) are players, and use Shapley values to distribute the terminal reward across generation steps. The method is evaluated on three tasks: sentiment control, text summarization, and instruction tuning, showing faster convergence and higher rewards compared to sparse RLHF and attention-based baselines.

**Strengths:**

The paper addresses an important problem in RLHF and provides a theoretically motivated approach based on cooperative game theory. The experimental design covers multiple tasks with different response lengths, demonstrating the method's applicability across various text generation scenarios. The writing is clear and the paper acknowledges some limitations in the appendix. The use of adaptive segmentation (token/span/sentence level) shows practical awareness of computational constraints.

**Weaknesses:**

The fundamental weakness lies in the mismatch between the cooperative game framework and autoregressive generation. Tokens are not independent agents that can form arbitrary coalitions; they are outputs of a sequential decision process with strong temporal dependencies. The characteristic function v(S) defined by space-filling is not theoretically justified and may not satisfy basic properties of cooperative games like superadditivity or monotonicity. The paper does not verify whether reward models can actually provide meaningful scores for such artificially constructed partial sequences.
The experimental choices are problematic for a 2025 submission. Using models from 2019-2020 (GPT-2, OpenLLaMA) instead of modern architectures (Llama 3, Qwen 2.5, Gemma 2) limits the relevance of findings. The absence of code generation and mathematical reasoning tasks is particularly concerning, since these domains represent critical RLHF applications where objective rewards (test pass rates, answer correctness) make credit assignment evaluation more reliable. The authors' explanation that "rule-based models" are unsuitable actually exposes a fundamental limitation: the method fails when tokens have strong causal dependencies, which is precisely where credit assignment is most needed.
The evaluation lacks rigor. Statistical significance is not tested despite marginal improvements. Win rates of 54.9% in LLM-as-judge evaluation are barely above random. Table 1 reports implausible standard deviations (0.00 for some methods), suggesting incomplete experimental runs.

**Questions:**

Could the authors provide evidence that reward models can meaningfully score partial sequences constructed with space-filling? Specifically, what is the distribution of scores for such sequences compared to naturally generated text, and how does this affect the validity of v(S) as a characteristic function?
Why were code generation and mathematical reasoning tasks excluded from evaluation? These tasks have objective rewards and strong causal dependencies, making them ideal test cases for credit assignment methods. Can the authors demonstrate SCAR's effectiveness on these tasks, or explicitly characterize the task properties where the method is expected to work versus fail?
The improvements over ABC are modest, and it's unclear whether they stem from Shapley values' fairness properties or simply from being a different dense reward scheme. Could the authors provide ablation studies comparing Shapley-based allocation against other principled dense reward distributions (e.g., based on gradient norms, attention entropy, or learned value functions)?
How does the method perform with modern language models (Llama 3, Qwen 2.5)? The use of GPT-2 and OpenLLaMA significantly limits the practical relevance of the findings in 2025.
Given that Owen value approximation is used for tractability, does Theorem 3.1's optimality guarantee still hold? What is the approximation error, and how does it affect the final policy quality?

---

> ### Author Response · Authors · 2025-11-26
> **Replies to Reviewer cak2 (1/2)**
>
> We sincerely thank the reviewer for the constructive and insightful comments. We provide clarifications in this response and have organized the concerns into the following items. Our replies to the comments and questions are presented below. The manuscript will be revised accordingly to reflect these updates.
>
> 1.**"Tokens are not independent agents that can form arbitrary coalitions; they are outputs of a sequential decision process with strong temporal dependencies. The characteristic function v(S) defined by space-filling is not theoretically justified and may not satisfy basic properties of cooperative games like superadditivity or monotonicity."**
>
> Response:  While Shapley value computation conceptually involves permutations of “player arrival’’ to ensure fairness, SCAR always preserves the original textual order of tokens when evaluating any coalition. We never shuffle token positions in the sequence passed to the RM. As a result, the RM consistently evaluates sequences that respect the causal dependencies and relative ordering produced by the LLM, even when certain intermediate tokens are masked or omitted within a given coalition.
>
> 2. **" The paper does not verify whether reward models can actually provide meaningful scores for such artificially constructed partial sequences. Could the authors provide evidence that reward models can meaningfully score partial sequences constructed with space-filling? Specifically, what is the distribution of scores for such sequences compared to naturally generated text, and how does this affect the validity of v(S) as a characteristic function?"**
>
> Response: We conducted a series of diagnostic evaluations (or sanity checks) to assess how well the reward models (RMs) handle partial or incomplete sequences. Representative examples include:
>
> Sentiment Control:
>
> response = "While some scenes were initially disturbing to sit through, they ultimately contributed to a deeply powerful and moving story."
> score = 9.9683
>
> text = [
>     "disturbing to sit through",
>     "deeply powerful",
>     "moving story",
>     "story",
>     "scenes",
>     "they ultimately contributed",
>     "moving",
>     "powerful",
>     "disturbing",
>     "While some scenes were initially to sit through, they ultimately contributed to a deeply powerful and moving story.",
>     "While some scenes were initially disturbing to sit through, they ultimately contributed to a deeply powerful and story.",
>     "While some scenes were initially disturbing to sit through, they ultimately contributed to a and moving story.",
> ]
>
> scores:
> [ -6.2960,  8.5214,  6.6705,  1.5888,  1.6383,  1.6652,  1.3947, 1.6845, -6.2683,  9.9356,  9.3892,  9.1769]
>
>
> Instruction-following:
>
> query = "Do you have any good video card suggestions for a good gaming PC?"
>
> response = """Let me see what my sources say. Hmm, interesting! It seems that the NVIDIA 1080 and the AMD RX 5700 have very positive reviews. However, the GTX 1080 is the top pick for the best GPU (Graphics Processing Unit), which is a kind of supercomputer that takes in your input and sends out a high-res image to your monitor.  It can do that very quickly. Let me know your budget or screen resolution, and I can suggest the best option for you."""
> score = 7.3948
>
> "Let me see what my sources say. Hmm, interesting!" 1.7638
> "It seems that the NVIDIA 1080 and the AMD RX 5700 have very positive reviews." 6.9662
> "However, the GTX 1080 is the top pick for the best GPU (Graphics Processing Unit), which is a kind of supercomputer that takes in your input and sends out a high-res image to your monitor." 4.5639
> "It can do that very quickly." 0.1931
> "Let me know your budget or screen resolution, and I can suggest the best option for you." 4.7314
>
> A few insights can be given based on the above examples:
>
> i. RMs provide reliable scores for partial sequences of moderate length.
> ii. They effectively identify semantically rich or task-relevant content. (e.g. "deeply powerful", "disturbing to sit through" for sentiment control, and GPU models for a query of video card suggestions).
> iii. Scores for very short sequences are less reliable. (e.g. the single positive words "moving" and "powerful" only got 1.3947 and 1.6845 scores). However, this is mitigated through the Shapley value mechanism, which averages marginal contributions across all possible sequence combinations, which include longer, more informative subsequences with more accurate scores.
>
> Therefore, the RM is reliable when evaluating long enough partial sequences. While RM evaluation may be noisy for extremely short fragments, this noise does not significantly affect SCAR’s overall performance.

---

> ### Author Response · Authors · 2025-11-26
> **Replies to Reviewer cak2 (2/2)**
>
> 3. **"The experimental choices are problematic for a 2025 submission. Using models from 2019-2020 (GPT-2, OpenLLaMA) instead of modern architectures (Llama 3, Qwen 2.5, Gemma 2) limits the relevance of findings. How does the method perform with modern language models (Llama 3, Qwen 2.5)?"**
>
> Response: Here we post some evaluations using recent models:
>
> Qwen2.5-3B on the instruction following task:
>
> |Method|	Performance|
> | -------- | ------- |
> |RLHF|	$6.84 \pm 0.01$|
> |ABC	|$5.41 \pm 0.00$|
> |SCAR-span	|$7.71 \pm 0.01$|
> |SCAR-sentence|	$7.56 \pm 0.01$|
>
> and Qwen2.5-7B on the instruction following task:
>
> |Method|	Performance|
> | -------- | ------- |
> |RLHF|	$6.53 \pm 0.23$|
> |ABC	|$6.78 \pm 0.17$|
> |SCAR-sentence	|$7.92 \pm 0.19$|
>
> 4. **" The absence of code generation and mathematical reasoning tasks is particularly concerning, since these domains represent critical RLHF applications where objective rewards (test pass rates, answer correctness) make credit assignment evaluation more reliable. These tasks have objective rewards and strong causal dependencies, making them ideal test cases for credit assignment methods. "**
>
> Response: SCAR assumes that the reward model can meaningfully evaluate long partial sequences. (See response 2)
>
> However, code and math tasks inherently break this assumption because partial outputs are unevaluable, discontinuous, and semantically brittle, making them poor testbeds for SCAR. Partial sequences in code/math are usually nonsensical from the perspective of objective evaluation. A single missing symbol or small arithmetic error can make the entire partial sequence completely worthless to the reward model.
>
> Therefore, these tasks would yield invalid or misleading evaluations of SCAR’s credit-assignment mechanism.
>
> Thank you again for taking the time to review our work. We hope that our responses have addressed your concerns, and we look forward to the possibility of a revised evaluation based on the clarifications provided.

---

> ### Comment · Reviewer_cak2 · 2025-11-26
>
> Thanks for the detailed response. While some of my concerns have been partially addressed, I remain unconvinced by the justification for excluding math and code tasks.
> The authors state that "code and math tasks inherently break this assumption because partial outputs are unevaluable." This misses the point entirely. Modern RL training on these tasks does not require evaluating partial sequences. DeepScaleR trained a 1.5B model to surpass O1-Preview on AIME using simple binary outcome rewards. The 1-shot RLVR work improved Qwen2.5-Math-1.5B from 36% to 73.6% on MATH500 with just one training example and 0/1 rewards. CoCoS achieved 35.8% gains on MBPP with 1B models. DeepSeek-R1 uses rule-based accuracy rewards from compilers and answer verifiers. None of these evaluate partial sequences — they simply check if the final answer is correct.
> If SCAR provides principled credit assignment via Shapley values, it should answer: given a complete response that received reward 1 or 0, which spans contributed to this outcome? This is precisely what Shapley values compute. The claim that these tasks are "poor testbeds" actually exposes a fundamental limitation of the method. Most dense reward research specifically targets math and code because these have objective evaluation criteria where credit assignment truly matters.
>
> I am willing to raise my score if the authors can provide even preliminary results on GSM8K or MBPP with outcome rewards. Otherwise, I must maintain that this limitation significantly reduces the contribution.

---

> ### Author Response · Authors · 2025-12-04
> **Replies to Reviewer cak2**
>
> Thanks for the further discussion. When we say that “code and math tasks inherently break this assumption because partial outputs are unevaluable,” we do not mean that RLHF is inapplicable to these tasks. Our point is that such tasks may be a poor setting for SCAR, because SCAR depends on accurate evaluations of (rather long) partial sequences. Nevertheless, we conducted some experiments with Qwen2.5-3B on the GSM8K dataset, and the preliminary results are as follows:
>
> |Method|Performance(\%)|
> | -------- | ------- |
> |RLHF|$85.4 \pm 0.3$|
> |SCAR|$87.3 \pm 0.2$|

---

### Official Review · Reviewer_BuRD · 2025-10-28

**Soundness:** 2
**Presentation:** 3
**Contribution:** 2
**Rating:** 4
**Confidence:** 4

**Summary:**

The paper addresses credit assignment in RLHF for aligning LLMs. It proposes SCAR, a method that uses Shapley-value-based attribution to assign token- or span-level contributions to a holistic reward. Experiments on IMDb TL;DR summarization and the Anthropic Helpful–Harmless (HH) dataset suggest effectiveness relative to baselines.

**Strengths:**

1. Introduces a principled, game-theoretic approach to reward attribution that is potentially more interpretable than heuristic token credit schemes.

2. Provides token/span-level granularity, which could help better and fast convergence.

3. Presents a clear, modular framework that can, be combined with standard RLHF pipelines.

**Weaknesses:**

1.	While Shapley values do not require independence per se, they assume a well-defined utility for any coalition of “players.” In LLMs, tokens are highly dependent and causal; the contribution of a token is context-sensitive and may change as later tokens are generated.
The paper should discuss how SCAR accounts for this sequential dependence beyond treating tokens as interchangeable players.

2.	Exact Shapley computation requires evaluating the utility for arbitrary coalitions. Using the reward model as the utility function directly may lead to out-of-distribution inputs the reward model was never trained to score. This raises questions about calibration, faithfulness, and bias in marginal contribution estimates.

3.	The adoption of the Owen approximation (grouped Shapley) is not justified with a principled grouping scheme. The paper lacks theoretical or empirical guidance on how to partition sequences into groups, how sensitive results are to the grouping strategy, and whether groups align with linguistic or structural units.

4.	Experiments are limited to GPT-2 and relatively simple scenarios. The absence of larger models, diverse tasks, and stronger baselines makes it difficult to assess generality and practical impact.

5.	Shapley-based methods can be expensive. The paper should quantify sampling budgets, variance, convergence, and training overhead relative to standard RLHF.

**Questions:**

1. How do you validate that the reward model is a faithful utility proxy for arbitrary coalitions of tokens or spans? Are there calibration studies or sanity checks?

2. What is the grouping rationale for the Owen approximation? How are groups defined (e.g., sentences, clauses, semantic units), and how sensitive are results to different grouping heuristics?

3. What is the computational overhead versus standard RLHF? How does this scale with sequence length?

4. How robust are attributions across different base models and domains? Have you tested SCAR on larger model families (e.g., Qwen, LLaMA), code or math tasks?

5. Do SCAR attributions reduce known reward hacking patterns?

6. How stable are token/span attributions across random seeds, prompt variations, and paraphrases? Is there high variance that could undermine their reliability for training?

---

> ### Author Response · Authors · 2025-11-19
> **Replies to Reviewer BuRD (1/4)**
>
> We greatly appreciate your valuable comments and suggestions, and we will revise the paper to reflect them.
>
> We have organized your concerns into the following items. The responses to your comments and questions are presented below.
>
> 1. **"While Shapley values do not require independence per se, they assume a well-defined utility for any coalition of “players.” In LLMs, tokens are highly dependent and causal; the contribution of a token is context-sensitive and may change as later tokens are generated. The paper should discuss how SCAR accounts for this sequential dependence beyond treating tokens as interchangeable players."**
>
> Response: While the Shapley value calculation mathematically involves iterating through permutations of ``player arrival'' to ensure fairness, SCAR strictly preserves the textual order of tokens during the evaluation of any coalition. We do not shuffle the positions of the tokens within the text passed to the RM. This ensures that the RM always evaluates a sequence that respects the causal dependencies and relative positions generated by the LLM, even if some intermediate tokens are masked/missing in a specific coalition.
>
> The ``context-sensitivity'' you mentioned—where an early token’s value depends on later tokens—is exactly what the Shapley value is designed to capture in cooperative game theory. By calculating the average marginal contribution of a token across all possible coalitions (including those that contain future tokens), SCAR assigns credit based on how that token interacts with the rest of the sequence.
> If an early token $t_i$ is essential for a later token $t_k$ to yield a high reward, coalitions containing ${t_i, t_k}$ will have high value, while those with only ${t_k}$ (without its necessary context $t_i$) will have low value. The Shapley calculation aggregates these interactions, effectively crediting $t_i$ for enabling the success of the later sequence.

---

> ### Author Response · Authors · 2025-11-19
> **Replies to Reviewer BuRD (2/4)**
>
> 2. **"Exact Shapley computation requires evaluating the utility for arbitrary coalitions. Using the reward model as the utility function directly may lead to out-of-distribution inputs that the reward model was never trained to score. This raises questions about calibration, faithfulness, and bias in marginal contribution estimates. How do you validate that the reward model is a faithful utility proxy for arbitrary coalitions of tokens or spans? Are there calibration studies or sanity checks?"**
>
> Response: We conducted a series of diagnostic evaluations (or sanity checks) to assess how well the reward models (RMs) handle partial or incomplete sequences. Representative examples include:
>
> Sentiment Control:
>
> response = "While some scenes were initially disturbing to sit through, they ultimately contributed to a deeply powerful and moving story."
> score = 9.9683
>
> text = [
>     "disturbing to sit through",
>     "deeply powerful",
>     "moving story",
>     "story",
>     "scenes",
>     "they ultimately contributed",
>     "moving",
>     "powerful",
>     "disturbing",
>     "While some scenes were initially to sit through, they ultimately contributed to a deeply powerful and moving story.",
>     "While some scenes were initially disturbing to sit through, they ultimately contributed to a deeply powerful and story.",
>     "While some scenes were initially disturbing to sit through, they ultimately contributed to a and moving story.",
> ]
>
> scores:
> [ -6.2960,  8.5214,  6.6705,  1.5888,  1.6383,  1.6652,  1.3947, 1.6845, -6.2683,  9.9356,  9.3892,  9.1769]
>
>
> Instruction-following:
>
> query = "Do you have any good video card suggestions for a good gaming PC?"
>
> response = """Let me see what my sources say. Hmm, interesting! It seems that the NVIDIA 1080 and the AMD RX 5700 have very positive reviews. However, the GTX 1080 is the top pick for the best GPU (Graphics Processing Unit), which is a kind of supercomputer that takes in your input and sends out a high-res image to your monitor.  It can do that very quickly. Let me know your budget or screen resolution, and I can suggest the best option for you."""
> score = 7.3948
>
> "Let me see what my sources say. Hmm, interesting!" 1.7638
> "It seems that the NVIDIA 1080 and the AMD RX 5700 have very positive reviews." 6.9662
> "However, the GTX 1080 is the top pick for the best GPU (Graphics Processing Unit), which is a kind of supercomputer that takes in your input and sends out a high-res image to your monitor." 4.5639
> "It can do that very quickly." 0.1931
> "Let me know your budget or screen resolution, and I can suggest the best option for you." 4.7314
>
> A few insights can be given based on the above examples:
>
> i. RMs provide reliable scores for partial sequences of moderate length.
>
> ii. They effectively identify semantically rich or task-relevant content. (e.g. "deeply powerful", "disturbing to sit through" for sentiment control, and GPU models for a query of video card suggestions).
>
> iii. Scores for very short sequences are less reliable. (e.g. the single positive words "moving" and "powerful" only got 1.3947 and 1.6845 scores). However, this is mitigated through the Shapley value mechanism, which averages marginal contributions across all possible sequence combinations, which include longer, more informative subsequences with more accurate scores.
>
> Therefore, the RM is reliable when evaluating long enough partial sequences. While RM evaluation may be noisy for extremely short fragments, this noise does not significantly affect SCAR’s overall performance.

---

> ### Author Response · Authors · 2025-11-19
> **Replies to Reviewer BuRD (3/4)**
>
> 3. **"The adoption of the Owen approximation (grouped Shapley) is not justified with a principled grouping scheme. The paper lacks theoretical or empirical guidance on how to partition sequences into groups, how sensitive results are to the grouping strategy, and whether groups align with linguistic or structural units. What is the grouping rationale for the Owen approximation? How are groups defined (e.g., sentences, clauses, semantic units), and how sensitive are results to different grouping heuristics?"**
>
> Response: Empirically, we recommend selecting the optimal granularity based on the nature of the task. Specifically, granularity should align with the length of meaningful units in the task:
>
> Sentiment control: 1-2 word/phrase segments (e.g., "good", "bad").
>
> Summarization: 5-10 word phrases containing core information (e.g., "6 meters long and 3 meters in width").
>
> Instruction tuning: 10+ word sentences (e.g., "Set clear boundaries: Establish clear boundaries around when you are and are not able to be distracted...").
>
> Empirically, a segmentation length of approximately 1/10 of the response length offers a good balance between performance and computational efficiency. This heuristic was consistently effective across tasks in our experiments.
>
> We also tested robustness to imperfect segmentation by using fixed-length (non-adaptive) splits. These led to only minor, non-significant drops in performance (still better than ABC and RLHF), suggesting SCAR is not overly sensitive to segmentation errors.
>
> For much longer generated sequences (e.g., 4k+ tokens), paragraph-level segmentation may be effective without significantly affecting computational efficiency.
>
> A comparative ablation study across token-level, span-level, and sentence-level, including both computational overhead and performance comparisons, is shown below:
> |Task	|Granularity|	Rewards	|Wall time usage (compared to vanilla RLHF)|
> | -------- | ------- | ------- | -------: |
> |Summarization|	token-level|	$4.33 \pm 0.15$	|8.3x|
> |Summarization|	span-level|	$4.35 \pm 0.11$	|1.06x|
> |Summarization|	sentence-level|	$4.16 \pm 0.08$	|1.02x|
> |Instruction-following|	token-level|	$7.32 \pm 0.01$|	131.9x|
> |Instruction-following|	span-level|	$7.29 \pm 0.02$|	1.09x|
> |Instruction-following|	sentence-level|	$7.31 \pm 0.01$|	1.05x|
>
> 4. **"Experiments are limited to GPT-2 and relatively simple scenarios. The absence of larger models, diverse tasks, and stronger baselines makes it difficult to assess generality and practical impact. How robust are attributions across different base models and domains? Have you tested SCAR on larger model families (e.g., Qwen, LLaMA), code or math tasks?"**
>
> Response: Here we present our results based on a 3B model (Qwen2.5-3B) on the instruction following task:
>
> |Method|	Performance|
> | -------- | ------- |
> |RLHF|	$6.84 \pm 0.01$|
> |ABC	|$5.41 \pm 0.00$|
> |SCAR-span	|$7.71 \pm 0.01$|
> |SCAR-sentence|	$7.56 \pm 0.01$|
>
> and Qwen2.5-7B with RLHF, ABC and SCAR on the instruction following task:
>
> |Method|	Performance|
> | -------- | ------- |
> |RLHF|	$6.53 \pm 0.23$|
> |ABC	|$6.78 \pm 0.17$|
> |SCAR-sentence	|$7.92 \pm 0.19$|
>
> 5. **"Shapley-based methods can be expensive. The paper should quantify sampling budgets, variance, convergence, and training overhead relative to standard RLHF. What is the computational overhead versus standard RLHF? How does this scale with sequence length?"**
>
>
> Response: Regarding computational efficiency, our empirical measurements show that the span-based variant of SCAR required only 1.06× to 1.09× the training time of vanilla RLHF, while the sentence-level variant incurred a smaller overhead of just 1.02× to 1.05×. This indicates that Shapley value computations in these variants contribute less than 10\% of total training time. We will add these in the revised manuscript.
>
> More broadly, our experiments have demonstrated that when the average segment length is about 1/10 of the full sequence length, the cost of computing Shapley values becomes negligible—yet the method still yields measurable improvements in performance.
>
> 6. **"Shapley-based methods can be expensive. The paper should quantify sampling budgets, variance, convergence, and training overhead relative to standard RLHF."**
>
> Response: This could be found in Figure 4 and previous responses.

---

> ### Author Response · Authors · 2025-11-19
> **Replies to Reviewer BuRD (4/4)**
>
> 7. **"Do SCAR attributions reduce known reward hacking patterns?"**
>
> Response: Unlike heuristic or separately trained dense reward methods (which can inadvertently bias the agent toward suboptimal behaviors or "hacks'' by altering the objective), SCAR is grounded in Potential-Based Reward Shaping (PBRS).
> Theorem 3.1 proves that SCAR preserves the optimal policy of the original sparse reward environment. Theoretically, this means SCAR does not introduce new global optima that correspond to ``hacks''; rather, it smooths the optimization landscape to help the agent locate the true optimal policy of the original RM more efficiently. While SCAR cannot fix a fundamentally flawed RM (i.e., if the RM itself encodes a hack), its attribution mechanism ensures the optimization process itself does not introduce additional instability or exploitation artifacts common in sparse reward settings.
>
> 8. **"How stable are token/span attributions across random seeds, prompt variations, and paraphrases? Is there high variance that could undermine their reliability for training?"**
>
> Response: During our experiments, we found that our method is not sensitive to random seeds, prompt variations, and paraphrases.
>
> Thank you again for taking the time to review our work. We hope that our responses have addressed your concerns. If you have any additional questions or concerns regarding our paper, please feel free to reach out. We look forward to the possibility of a revised evaluation based on the clarifications provided.

---

### Official Review · Reviewer_UrLU · 2025-10-29

**Soundness:** 3
**Presentation:** 3
**Contribution:** 2
**Rating:** 6
**Confidence:** 4

**Summary:**

This paper proposes SCAR, a method based on the Shapley value in cooperative game theory, to address the reward sparsity problem in RLHF. SCAR fairly allocates sequence-level rewards to tokens or text segments (such as phrases and sentences) in the generated text, thereby providing dense reward signals without the need for additional training of a critique model or fine-grained human annotations. The main contributions of the paper are as follows:
Proposing a Shapley value-based credit assignment method with theoretical guarantees (e.g., efficiency, symmetry, linearity);
Reducing computational complexity through adaptive text segmentation (token-level, phrase-level, sentence-level) and Owen value approximation;
Theoretically proving that SCAR preserves the optimal policy (based on Potential-Based Reward Shaping);
Conducting empirical validation on three tasks—sentiment control, text summarization, and instruction tuning—showing that SCAR achieves faster convergence and higher final rewards compared to the standard RLHF, Uniform, and Attention-Based Credit (ABC) baselines.

**Strengths:**

Originality: Introducing the Shapley value into RLHF credit assignment represents an innovative integration of cooperative game theory and RLHF, distinct from heuristic methods that rely on attention weights (e.g., ABC).
Quality: The method is rigorously designed, encompassing theoretical analysis (policy invariance), efficient approximations (Owen value, adaptive segmentation), and comprehensive experiments (three tasks, multiple baselines, and statistical validation).
Clarity: The paper is well-structured with detailed descriptions of the method; figures and tables effectively aid understanding, and the appendix provides hyperparameters, proofs, and additional experiments.
Significance: It addresses the core issue in RLHF—credit assignment—can enhance the efficiency and stability of LLM alignment, and holds broad application potential.

**Weaknesses:**

Computational overhead: Despite reducing complexity through the use of Owen values and adaptive segmentation, Shapley value approximation still introduces significant computational costs (e.g., Token-level SCAR requires 48 GPU hours for the summarization task, compared to 7 hours for Span-level). It may remain impractical for extremely long sequences or large-scale LLMs.
Reward model assumptions: SCAR assumes that the reward model can provide meaningful scoring for partial sequences (Eq. 3). However, some reward models (such as rule-based models or those that only evaluate final answers) may not be applicable, limiting its generalizability.
Limitations in experimental scale: The experiments are based on smaller models (e.g., GPT-2 small, Pythia-1B) and have not been validated on larger LLMs (e.g., LLaMA-2 or GPT-3 scale), which may affect the generalizability of the conclusions.
Hyperparameter sensitivity: The selection of α (Shapley reward coefficient) may affect performance, but the paper does not provide a systematic sensitivity analysis or guidelines for its selection.

**Questions:**

Computational efficiency: What is the computational cost of SCAR for tasks involving generating long documents? Are there plans to develop more efficient approximation algorithms ?
Generalizability of the reward model: How would SCAR be adjusted if the reward model cannot reliably evaluate partial sequences? Has consideration been given to using generative reward models or multi-step evaluation?
Scalability: Are there plans to validate SCAR on larger-scale LLMs or more diverse tasks?
Hyperparameter optimization: Is the selection of α and segmentation strategies (token/phrase/sentence level) task-dependent? Can guidelines for the automatic selection of these hyperparameters be provided?
Comparison with contemporaneous work: What are the relative advantages of SCAR in terms of efficiency and effectiveness compared to the method by Koo et al. (2025)? Has a direct comparison been conducted?

---

> ### Author Response · Authors · 2025-11-24
> **Replies to Reviewer UrLU (1/2)**
>
> **Response to Question 1: Computational Efficiency**
>
> Thank you for raising this important point regarding the computational cost of SCAR. We address computational efficiency through two primary mechanisms described in **Section 3.4**:
>
> 1.  **Approximation via Owen Values:** To avoid the exponential complexity ($O(2^N)$) of exact Shapley value calculation, we utilize Owen values. This reduces the complexity to quadratic in the number of text units ($N$), making the approach tractable for standard training pipelines.
> 2.  **Adaptive Text Segmentation:** As detailed in **Section 3.4** ("Adaptive Text Segmentation as Players"), we adapt the granularity of the "players" in our cooperative game based on the response length. By grouping tokens into larger syntactic structures—such as spans (via constituency parsing) or sentences—we drastically reduce $N$. Therefore, for tasks involving very long documents, we can keep the number of players manageable regardless of total token count.
>
> Regarding future work, as noted in **Appendix A (Limitations)**, we plan to explore even more efficient approximation techniques to further minimize overhead for extremely large context windows.
>
> **Response to Question 2: Generalizability of the Reward Model**
>
> This is an insightful question. SCAR currently relies on the Reward Model (RM) to provide a scalar score for partial sequences (Equation 3), treating the formed coalitions as "complete" inputs. If the RM cannot reliably score partial sequences (e.g., a math verifier that only checks the final answer and returns zero for anything else), SCAR would indeed require adjustment. A potential solution, which we briefly discuss in **Section 3.2** ("Characteristic Function"), is to sample from the language model to fill in the masked or dropped parts corresponding to players not in the coalition ($u_j \notin S$). Instead of evaluating the partial text directly, we would estimate the expected reward by keeping the coalition tokens fixed and generating the rest of the sequence using the current policy. While this approach would allow SCAR to function with RMs that require full, coherent contexts (like mathematical reasoning models), it would introduce additional computational cost compared to our current implementation, which uses empty/padding tokens for efficiency.
>
> **Response to Question 3: Scalability**
>
> Yes, validating SCAR on larger scales is a priority.
>
> *   **Current Validation:** Our current experiments demonstrate effectiveness on models ranging from **1B (Pythia)** to **7B (OpenLLaMA)** parameters across tasks with varying complexity (Sentiment Control, Summarization, and Instruction Tuning).
> *   **Future Plans:** As explicitly stated in **Appendix A**, a key direction for future work is the "rigorous evaluation on larger-scale language models and broader tasks." We aim to verify that the efficiency gains from our adaptive segmentation strategy hold—and potentially become even more critical—when applied to larger architectures (e.g., 70B+ models) where the base generation cost is significantly higher.
>
> **Response to Question 4: Hyperparameter Optimization**
>
> You are correct that $\alpha$ and the segmentation strategy are task-dependent. Based on our empirical findings (detailed in **Appendix D** and **Section 3.4**), we can provide the following guidelines:
>
> 1.  **Segmentation Strategy:** This should be selected based on the expected output length to balance granularity and speed.
>     *   *Short outputs (<50 tokens):* **Token-level** (e.g., Sentiment Control).
>     *   *Medium outputs (50-200 tokens):* **Span-level** (e.g., Summarization).
>     *   *Long outputs (>200 tokens):* **Sentence-level** (e.g., long-form generation).
> 2.  **Alpha ($\alpha$):** We found that a high coefficient for the Shapley reward is generally robust. In our experiments, we used $\alpha=0.8$ for sentiment/instruction tuning and $\alpha=1.0$ for summarization. Because SCAR is theoretically grounded to preserve the optimal policy (**Theorem 3.1**), relying heavily on the dense signal does not introduce the bias often associated with heuristic shaping, allowing for higher $\alpha$ values.

---

> ### Author Response · Authors · 2025-11-24
> **Replies to Reviewer UrLU (2/2)**
>
> **Response to Question 5: Comparison with Contemporaneous Work (Koo et al., 2025)**
>
> We discuss the differences between SCAR and Koo et al. (2025) in **Section 2.3**. The relative advantages of SCAR lie in its independence from auxiliary supervision and its computational simplicity:
>
> 1.  **No Assumption of Intermediate Rewards:** A critical advantage of SCAR is that it does not require a powerful auxiliary language model or oracle to provide intermediate step rewards. Koo et al. (2025) typically assume the existence of such a model (or equivalent interpretability signals) to estimate per-token rewards. In contrast, **SCAR extracts dense rewards purely from the standard, sparse scalar reward model** using game-theoretic principles. This makes SCAR applicable in standard RLHF settings where only a final reward model is available.
> 2.  **Optimization Framework:** Koo et al. employ a bilevel optimization framework (using Bayesian Optimization) to learn shaping parameters, which introduces an outer optimization loop. SCAR calculates the dense reward signal directly via the Owen value approximation (Equation 6) without needing to learn shaping parameters, offering a more streamlined integration into PPO pipelines.
> 3.  **Handling Long Sequences:** SCAR introduces **Adaptive Text Segmentation** (Section 3.4), whereas Koo et al. focus on token-level attribution. Our approach allows for significantly better efficiency on longer sequences where token-level calculations become prohibitive.
>
> Nevertheless, here we post the direct comparison:
>
> |Task	|Method   |Rewards	|
> | -------- | ------- | ------- |
> |Instruction-following|OpenLLaMA-7B+RLHF|	$6.93 \pm 0.00$|
> |Instruction-following|OpenLLaMA-7B+BO(SHAP+LIME)|	$6.57 \pm 0.05$|
> |Instruction-following|OpenLLaMA-7B+SCAR|	$7.31 \pm 0.01$|
>
> Thank you again for taking the time to review our work. We hope that our responses have addressed your concerns, and we look forward to the possibility of a revised evaluation based on the clarifications provided.

---

### Official Review · Reviewer_N6FG · 2025-11-01

**Soundness:** 3
**Presentation:** 3
**Contribution:** 2
**Rating:** 6
**Confidence:** 3

**Summary:**

SCAR proposes to use Shapely Values to get a dense token/span-level reward for RLHF. The properties of Shapely values enable SCAR to preserve the original optimal policy and facilitate faster convergence. The method further uses Owen value approximation to make the computation tractable. The evaluations are carried out on Sentiment Control, Summarization and Instruction tuning tasks

**Strengths:**

1.	The use of Shapley values to redistribute sparse rewards into dense rewards has a strong theoretical foundation. Further, the dense learning signal does not change the optimal policy.
2.	Adaptive segmentation and Owen value approximations make the approach tractable and suited for real-world use.
3.	Experiments show that SCAR achieves faster convergence and preference compared to the baselines.

**Weaknesses:**

1.	The proposed method focuses on a novel way to convert sparse reward into dense reward(s) and the experiments demonstrate the efficacy of dense rewards in PPO training. However, there are other RL methods, e.g. OREO (Wang et al., 2025)  and DQO (Liu et al., 2024), that explicitly formulate intermediate steps as an MDP and employ soft Q-learning to learn the policy. These methods can also use process level supervision.  A major concern with the experiment section is that they focus only on improving the learnability of PPO. Maybe they should also demonstrate that other methods such as DQO can also be improved using the proposed sparse to dense reward conversion.

             a. Wang, Chaojie, et al. "Q*: Improving multi-step reasoning for llms with deliberative planning", 2024
             b. Liu et al. et al. "Enhancing multi-step reasoning abilities of language models through direct q-function optimization." 2024.


2.	The comparisons with ABC is appropriate; however, it would strengthen the paper to include evaluations against more recent dense-reward methods, such as critic-based auxiliary models, learned dense reward models, and the contemporary works cited. While SCAR has advantages (e.g., no need for dense human annotations or training a separate reward model), comparing its performance with these methods is essential to contextualize its effectiveness. Some potential baselines to consider are the following:

         a. Chan, Alex J., et al. "Dense reward for free in reinforcement learning from human feedback." arXiv preprint arXiv:2402.00782 (2024).
         b. Yoon, Eunseop, et al. "Tlcr: Token-level continuous reward for fine-grained reinforcement learning from human feedback."  ACL Fndings (2024).
         c. Koo, Ryan, et al. "Learning explainable dense reward shapes via bayesian optimization." arXiv preprint arXiv:2504.16272 (2025).


3.	The reward model (RM) score alone provides limited insight into the model’s actual improvement. Including a human evaluation on at least a subset of the test data would have helped assess how well the reward model’s judgments align with human preferences.


4.	A qualitative comparison of model outputs would be valuable to illustrate the specific kinds of improvements or additional learning that the dense reward model provides over the baseline methods.

**Questions:**

Figure 4 indicates that for the summarization task, token-level dense rewards underperform compared to span-level dense rewards. It would be useful for the authors to discuss why this happens. Does it suggest that an optimal reward granularity exists—somewhere between token-level and sequence-level—to balance precision and contextual coherence? An ablation study varying the span sizes used for dense rewards could clarify this relationship and reveal how reward granularity affects learning stability and performance.

---

> ### Author Response · Authors · 2025-11-24
> **Replies to Reviewer N6FG (1/3)**
>
> We sincerely thank the reviewer for the constructive and insightful comments. We provide clarifications to the reviewer within this response. We have organized your concerns into the following items. The responses to your comments and questions are presented below.
>
> 1. **"Maybe they should also demonstrate that other methods such as DQO can also be improved using the proposed sparse to dense reward conversion."**
>
> Response: Thank you for pointing this out. Evaluating SCAR with additional RL methods could indeed provide stronger evidence. We have also experimented with DQO using Qwen2.5 on our tasks:
>
> |Task	|Method   |Rewards	|
> | -------- | ------- | ------- |
> |Summarization|	Qwen2.5-3B+DQO|	$2.14 \pm 0.19$	|
> |Summarization|	Qwen2.5-3B+DQO+SCAR|	$4.40 \pm 0.15$	|
> |Instruction-following|Qwen2.5-3B+DQO|	$6.72 \pm 0.22$|
> |Instruction-following|Qwen2.5-3B+DQO+SCAR|	$7.23 \pm 0.06$|
>
> The results above indicate that SCAR can enhance RL methods beyond PPO on these general tasks. We will include this in the next version of the paper.
>
> 2. **"It would strengthen the paper to include evaluations against more recent dense-reward methods, such as critic-based auxiliary models, learned dense reward models, and the contemporary works cited."**
>
> Response: Our method, SCAR, obtains dense rewards without requiring additional models. In contrast, TLCR necessitates an additional discriminator to compute dense rewards. This introduces a level of inherent complexity in TLCR that SCAR does not have. A direct comparison between the two would therefore obscure substantial differences in complexity and resource requirements. For this reason, we prefer to compare SCAR with "free" dense reward methods, among which ABC is the only method that meets this criterion. Regarding the other two cited works, “Dense reward for free in reinforcement learning from human feedback.” corresponds directly to ABC, and “Learning explainable dense reward shapes via Bayesian optimization.” reports performance that is substantially worse than ours on the same task:
>
> |Task	|Method   |Rewards	|
> | -------- | ------- | ------- |
> |Instruction-following|OpenLLaMA-7B+RLHF|	$6.93 \pm 0.00$|
> |Instruction-following|OpenLLaMA-7B+BO(SHAP+LIME)|	$6.57 \pm 0.05$|
> |Instruction-following|OpenLLaMA-7B+SCAR|	$7.31 \pm 0.01$|
>
> 3. **"The reward model (RM) score alone provides limited insight into the model’s actual improvement. Including a human evaluation on at least a subset of the test data would have helped assess how well the reward model’s judgments align with human preferences."**
>
> Response: Table 2 shows the results where we use gemini-2.5-pro to compare anonymized model outputs (SCARvs.baselines) on coherence, relevance, conciseness, and overall quality. This comparison offers insight into the extent to which the reward model’s judgments are consistent with human-aligned preferences and provides further validation of SCAR’s effectiveness.

---

> ### Author Response · Authors · 2025-11-24
> **Replies to Reviewer N6FG (2/3)**
>
> 4. **"A qualitative comparison of model outputs would be valuable to illustrate the specific kinds of improvements or additional learning that the dense reward model provides over the baseline methods."**
>
> Responses: Some examples are as follows:
> Sentiment Control:
>
> response = "While some scenes were initially disturbing to sit through, they ultimately contributed to a deeply powerful and moving story."
> score = 9.9683
>
> text = [
>     "disturbing to sit through",
>     "deeply powerful",
>     "moving story",
>     "story",
>     "scenes",
>     "they ultimately contributed",
>     "moving",
>     "powerful",
>     "disturbing",
>     "While some scenes were initially to sit through, they ultimately contributed to a deeply powerful and moving story.",
>     "While some scenes were initially disturbing to sit through, they ultimately contributed to a deeply powerful and story.",
>     "While some scenes were initially disturbing to sit through, they ultimately contributed to a and moving story.",
> ]
>
> scores:
> [ -6.2960,  8.5214,  6.6705,  1.5888,  1.6383,  1.6652,  1.3947, 1.6845, -6.2683,  9.9356,  9.3892,  9.1769]
>
>
> Instruction-following:
>
> query = "Do you have any good video card suggestions for a good gaming PC?"
>
> response = """Let me see what my sources say. Hmm, interesting! It seems that the NVIDIA 1080 and the AMD RX 5700 have very positive reviews. However, the GTX 1080 is the top pick for the best GPU (Graphics Processing Unit), which is a kind of supercomputer that takes in your input and sends out a high-res image to your monitor.  It can do that very quickly. Let me know your budget or screen resolution, and I can suggest the best option for you."""
> score = 7.3948
>
> "Let me see what my sources say. Hmm, interesting!" 1.7638
> "It seems that the NVIDIA 1080 and the AMD RX 5700 have very positive reviews." 6.9662
> "However, the GTX 1080 is the top pick for the best GPU (Graphics Processing Unit), which is a kind of supercomputer that takes in your input and sends out a high-res image to your monitor." 4.5639
> "It can do that very quickly." 0.1931
> "Let me know your budget or screen resolution, and I can suggest the best option for you." 4.7314
>
> The sampled examples demonstrate that SCAR can assign reasonable and well-differentiated dense rewards to each component.

---

> ### Author Response · Authors · 2025-11-24
> **Replies to Reviewer N6FG (3/3)**
>
> 5. **"Figure 4 indicates that for the summarization task, token-level dense rewards underperform compared to span-level dense rewards. It would be useful for the authors to discuss why this happens. Does it suggest that an optimal reward granularity exists—somewhere between token-level and sequence-level—to balance precision and contextual coherence? An ablation study varying the span sizes used for dense rewards could clarify this relationship and reveal how reward granularity affects learning stability and performance."**
>
> Response: To clarify, the x-axis in Figure 4 represents GPU hours. This indicates that, under the same time budget, token-level dense rewards cannot match the performance of span-level dense rewards. However, when computation efficiency is not taken into account, the two approaches exhibit similar performance (see Figure 5 in Appendix C).
>
> Regarding the choice of reward granularity, we recommend selecting the granularity according to the intrinsic structure of the task. In particular, the segmentation should align with the length of the task’s meaningful units:
>
> Sentiment control: 1-2 word/phrase segments (e.g., "good", "bad").
>
> Summarization: 5-10 word phrases containing core information (e.g., "6 meters long and 3 meters in width").
>
> Instruction tuning: 10+ word sentences (e.g., "Set clear boundaries: Establish clear boundaries around when you are and are not able to be distracted...").
>
> Empirically, we find that a segmentation length of approximately one-tenth of the response length strikes an effective balance between performance and computational efficiency. This heuristic worked consistently well across all tasks evaluated in our experiments.
>
> We also tested robustness to imperfect segmentation by using fixed-length (non-adaptive) splits. These led to only minor, non-significant drops in performance (still better than ABC and RLHF), suggesting SCAR is not overly sensitive to segmentation errors.
>
>
> A comparative ablation study across token-level, span-level, and sentence-level, including both computational overhead and performance comparisons, is shown below:
> |Task	|Granularity|	Rewards	|Wall time usage (compared to vanilla RLHF)|
> | -------- | ------- | ------- | -------: |
> |Summarization|	token-level|	$4.33 \pm 0.15$	|8.3x|
> |Summarization|	span-level|	$4.35 \pm 0.11$	|1.06x|
> |Summarization|	sentence-level|	$4.16 \pm 0.08$	|1.02x|
> |Instruction-following|	token-level|	$7.32 \pm 0.01$|	131.9x|
> |Instruction-following|	span-level|	$7.29 \pm 0.02$|	1.09x|
> |Instruction-following|	sentence-level|	$7.31 \pm 0.01$|	1.05x|
>
> Thank you again for taking the time to review our work. We hope that our responses have addressed your concerns, and we look forward to the possibility of a revised evaluation based on the clarifications provided.

---

### Meta-Review · Area_Chair_pMCF · 2026-01-08

**Summary:**

This paper introduces Shapley Values, a concept from Game theory, in converting sparse reward to dense reward in RLHF for LLM. The main idea is that Shapley Values provide a way to compute score for a given token/snippet in the generation using a formula that intuitively measures how often that token leads to a partial generation doing better. This is computed using the reward model on the partial generation and thus, therefore, require the reward model to be able to accurately compute score for partial generations. The proposed approach SCAR has a nice theoretically proper that it does not change the set of optimal policies. As the original formulation is combinatorially expensive, an approximation is computed using Owen values. Results are presented with old and small models like GPT-2 and notably math/coding tasks are omitted since authors find they are not suited for SCAR style partial reward evaluation.

The Shapley Value concept is a neat idea that this paper introduces, but I feel this paper is currently not ready for publication based on the following concerns:

- Results are presented using small models with two important LLM applications such as math and coding omitted. Authors have tried to answer this in rebuttal using some results but more is needed before this paper can be accepted. Using modern LLMs even if they are under 70B will be useful. And math and coding domains should be tried, even if the result is negative, as they are important applications.

- Conceptually, Shapley Values require reward model to be able to compute score of a partial subset of generation snippets which may not always be accurate given that reward models aren't trained this way. More experimental results can address this concerns specially on more domain.

**Reviewer Concerns:**

Reviewers raised the following concern:

1. Lack of experiments with newer models. Authors have experimented with old small models like GPT-2. In response to this concern, authors have provided additional result with Qwen 2.5 models. This partially addresses this concern, but this needs to be included more comprehensively.

2. Lack of experiments on math and coding. This remains unaddressed since authors mentioned that these domains aren't suited for SCAR. However, authors have provided a seed result where SCAR actually seems to help. I would recommend this complete result with all baselines and on more domains be included.

3. Computational complexity of performing SCAR: Computing Shapley Values is combinatorially expensive, but using Owen approximation the authors show that they can reduce it to quadratic which by using bigger snippets like span, leads to only a small increase in wall clock time (under 10% training time). I believe this concern to be addressed.

4. Comparison with related work. Authors have partially addressed these concerns.

**Reviewer Scores:**

1. Reviewer N6FG raised main concerns on related work and comparison with other work that densify sparse reward. Authors have partially addressed this by producing more results. Authors say that SCAR is not fair to compare with approaches that train more models, but I don't see why this is an issue since the main criterion should be final accuracy. If another approach uses more training to do better, it is preferable in many cases. If instead authors want to make a wall clock argument, then they should evaluate it on both time and accuracy and state this. Overall, I feel this score is best left at 6.

2. Reviewer UrLU raised main concerns on computational efficiency and scalability. On the former, the authors address it but the latter remains unaddressed. This score can be notched up a bit to 7.

3. Reviewer BuRD raised concerns on conceptual issues with Shapley values, computational complexity, scalability. The scalability concerns are unaddressed. I would increase their score to 5.

4. Reviewer cak2 raised concerns on suitability of reward on partial generation, use of old small LLMs and . These concerns at large remain unaddressed or partially addressed. I would therefore only increase the score to 3.

Overall, this paper explores a novel idea but is currently not ready for acceptance.

---

### Decision · Program_Chairs · 2026-01-26

Reject